# Genomic Risk Prediction for Breast Cancer in Older Women

**DOI:** 10.3390/cancers13143533

**Published:** 2021-07-14

**Authors:** Paul Lacaze, Andrew Bakshi, Moeen Riaz, Suzanne G. Orchard, Jane Tiller, Johannes T. Neumann, Prudence R. Carr, Amit D. Joshi, Yin Cao, Erica T. Warner, Alisa Manning, Tú Nguyen-Dumont, Melissa C. Southey, Roger L. Milne, Leslie Ford, Robert Sebra, Eric Schadt, Lucy Gately, Peter Gibbs, Bryony A. Thompson, Finlay A. Macrae, Paul James, Ingrid Winship, Catriona McLean, John R. Zalcberg, Robyn L. Woods, Andrew T. Chan, Anne M. Murray, John J. McNeil

**Affiliations:** 1Department of Epidemiology and Preventive Medicine, School of Public Health and Preventive Medicine, Monash University, Melbourne, VIC 3004, Australia; Andrew.Bakshi1@monash.edu (A.B.); Moeen.Riaz@monash.edu (M.R.); Suzanne.Orchard@monash.edu (S.G.O.); jane.tiller@monash.edu (J.T.); Johannes.Neumann@monash.edu (J.T.N.); Prue.Carr@monash.edu (P.R.C.); john.zalcberg@monash.edu (J.R.Z.); Robyn.Woods@monash.edu (R.L.W.); john.mcneil@monash.edu (J.J.M.); 2Clinical and Translational Epidemiology Unit, MGH Cancer Center, Massachusetts General Hospital and Harvard Medical School, Boston, MA 02108, USA; ADJOSHI@mgh.harvard.edu (A.D.J.); EWarner@mgh.harvard.edu (E.T.W.); AKMANNING@mgh.harvard.edu (A.M.); achan@mgh.harvard.edu (A.T.C.); 3Alvin J. Siteman Cancer Center, Division of Public Health Sciences, Department of Surgery, Washington University School of Medicine, St Louis, MO 63110, USA; yin.cao@wustl.edu; 4Precision Medicine, School of Clinical Sciences at Monash Health, Monash University, Melbourne, VIC 3168, Australia; Tu.Nguyen-Dumont@monash.edu (T.N.-D.); Melissa.Southey@monash.edu (M.C.S.); Roger.Milne@cancervic.org.au (R.L.M.); 5Department of Clinical Pathology, University of Melbourne, Melbourne, VIC 3010, Australia; 6Cancer Epidemiology Division, Cancer Council Victoria, Melbourne, VIC 3004, Australia; 7Division of Cancer Prevention, National Cancer Institute, Rockville, MD 20892, USA; fordl@mail.nih.gov; 8Department of Genetics and Genomic Sciences, Icahn School of Medicine at Mount Sinai, New York, NY 10029, USA; robert.sebra@mssm.edu (R.S.); eric.schadt@mssm.edu (E.S.); 9Personalised Oncology Division, Walter and Eliza Hall Institute Medical Research, Faculty of Medicine, University of Melbourne, Melbourne, VIC 3052, Australia; lucyjgately@hotmail.com (L.G.); Peter.Gibbs@mh.org.au (P.G.); 10Department of Genomic Medicine, Royal Melbourne Hospital, University of Melbourne, Parkville, Melbourne, VIC 3050, Australia; Bryony.Thompson@mh.org.au (B.A.T.); Finlay.Macrae@mh.org.au (F.A.M.); Paul.James@petermac.org (P.J.); ingrid.winship@mh.org.au (I.W.); 11Department of Anatomical Pathology, Alfred Hospital, Melbourne, VIC 3004, Australia; C.McLean@alfred.org.au; 12Berman Center for Outcomes and Clinical Research, Hennepin Healthcare Research Institute, Hennepin Healthcare, University of Minnesota, Minneapolis, MN 55404, USA; AMurray@bermancenter.org

**Keywords:** genomics, breast cancer, risk prediction, polygenic risk score, germline

## Abstract

**Simple Summary:**

We designed a study specifically to assess the performance of genomic risk prediction for breast cancer (BC) in older women aged ≥70 years. We assessed the effects of a polygenic risk score (PRS) for BC and rare pathogenic variants (PVs) in BC susceptibility genes, on incident BC risk in a prospective cohort of 6339 older women (mean age 75 years). During a median follow-up time of 4.7 years, the PRS was an independent predictor of incident BC risk, with women in the top quintile of the PRS distribution having over two-fold higher incident BC risk than women in the lowest quintile. Among 41 carriers of PVs in BC susceptibility genes, we observed no incident BC diagnoses. Our study demonstrates that a PRS still predicts incident BC risk in women aged 70 years and older, suggesting the potential clinical utility of the PRS extends to this older age group.

**Abstract:**

Genomic risk prediction models for breast cancer (BC) have been predominantly developed with data from women aged 40–69 years. Prospective studies of older women aged ≥70 years have been limited. We assessed the effect of a 313-variant polygenic risk score (PRS) for BC in 6339 older women aged ≥70 years (mean age 75 years) enrolled into the ASPREE trial, a randomized double-blind placebo-controlled clinical trial investigating the effect of daily 100 mg aspirin on disability-free survival. We evaluated incident BC diagnoses over a median follow-up time of 4.7 years. A multivariable Cox regression model including conventional BC risk factors was applied to prospective data, and re-evaluated after adding the PRS. We also assessed the association of rare pathogenic variants (PVs) in BC susceptibility genes (*BRCA1/BRCA2/PALB2/CHEK2/ATM*). The PRS, as a continuous variable, was an independent predictor of incident BC (hazard ratio (HR) per standard deviation (SD) = 1.4, 95% confidence interval (CI) 1.3–1.6) and hormone receptor (ER/PR)-positive disease (HR = 1.5 (CI 1.2–1.9)). Women in the top quintile of the PRS distribution had over two-fold higher risk of BC than women in the lowest quintile (HR = 2.2 (CI 1.2–3.9)). The concordance index of the model without the PRS was 0.62 (95% CI 0.56–0.68), which improved after addition of the PRS to 0.65 (95% CI 0.59–0.71). Among 41 (0.6%) carriers of PVs in BC susceptibility genes, we observed no incident BC diagnoses. Our study demonstrates that a PRS predicts incident BC risk in women aged 70 years and older, suggesting potential clinical utility extends to this older age group.

## 1. Introduction

Breast cancer (BC) risk prediction models may be improved by including genomic risk scores. A polygenic risk score (PRS) aggregates the effect of many common BC risk-associated variants into a single measure [1,2,3,4]. Common BC risk-associated genetic variants used in a PRS, together, are estimated to account for 18% of familial BC risk [5]. Predictive performance of a PRS for BC has mostly been assessed in women aged 40–69 years [1,3,6,7,8,9,10]. PRS performance, in terms of risk prediction in older women (aged ≥70 years), is unclear, despite a high proportion of BC diagnoses occurring in this age group. It is also unclear whether the predictive performance of a PRS for BC attenuates with age. Given the emerging clinical utility of PRS for BC risk prediction and stratification, this requires further assessment.

A small proportion of women (<5%) carry rare pathogenic variants (PVs) in BC predisposition genes, including *BRCA1*, *BRCA2*, *PALB2*, *CHEK2* and *ATM* [11]. Rare PVs can be detected by predictive clinical genetic testing and are of high clinical significance [12,13], predisposing women to BC earlier in life [6,7,8,11]. Rare PVs account for ~25% of familial BC risk [14]. Recent studies have suggested that the increase in BC risk associated with rare PVs in high-risk genes (e.g., *BRCA1*, *BRCA2*) may decrease with increasing age [6,8,10,12,13]. If rare PVs have age-dependent effects on BC risk, this may have important clinical implications for the appropriateness of offering predictive genetic testing to older people.

Many clinical studies have measured the association between rare PVs and BC risk [6,7,8,11]. More recently, risk conferred by a 313-variant PRS for BC has been measured in meta-analysis of ten studies, with no evidence of age-related attenuation in the predictive performance of the PRS reported [1]. The same PRS has been assessed across several subsequent PRS validation studies [2,3,4]. However, participant numbers aged ≥70 years were limited. Here, we evaluate the predictive performance of (i) a PRS for BC and (ii) rare PVs in a prospective cohort of 6339 women aged ≥70 years.

## 2. Materials and Methods

### 2.1. Study Sample

The study sample comprised female participants of the ASPirin in Reducing Events in the Elderly (ASPREE) trial—a randomized, placebo-controlled, clinical trial investigating the effect of daily 100mg aspirin on disability-free survival [15,16,17]. Study design [18,19], recruitment [20], and baseline characteristics [21] have been published previously. The trial recruited 19,114 individuals from Australia (*n* = 16,703) and the United States (*n* = 2411) aged 70 years or older (≥65 years for US ethnic minorities), who, at enrolment, were free from diagnosed cardiovascular disease events, dementia, physical disability and life-threatening cancer diagnoses. Biospecimens and consent for genetic analysis were obtained from 14,576 participants. The median follow-up time (randomization period) was 4.7 years (interquartile range 2.1 years). The study received approvals from the Alfred Hospital Research Ethics Committee (Project 390/15) and is registered (NCT01038583). All participants provided written informed consent for genetic research. The final analysis for this study was conducted on 6339 female ASPREE participants from Australia aged ≥70 years, for whom both genome-wide genotyping and targeted sequencing data were available.

### 2.2. Genome-Wide Genotyping and Polygenic Risk Score

DNA samples were genotyped using the Axiom 2.0 Precision Medicine Diversity Research Array (Thermo Fisher Scientific (TFS), Waltham, MA, USA) following standard protocols. Only participants with European genetic ancestry (>95% of female participants) were included, to mitigate population stratification bias. Genetic ancestry was defined using principal component analysis (PCA) based on the 1000 Genomes reference population, with participants outside of the Non-Finnish European ancestry cluster excluded (Appendix A) [22]. Imputation was performed using the TopMED Server (European samples) [23]. After variants with low imputation quality scores (*r*^2^ < 0.3) were excluded, a PRS was calculated based on the 313-variant score previously described [1], using genotypes for the remaining 271 variants (for a list of the 313 variants in the PRS, indicating the 271 variants included in the final analysis, see Appendix A). Using Plink version 1.9, we calculated the PRS for each individual as the weighted sum of the effect size for the number of risk alleles at each variant [1].

### 2.3. Targeted Gene Panel Sequencing

Our custom gene panel [24] included BC predisposition genes that are incorporated into the Breast and Ovarian Analysis of Disease Incidence and Carrier Estimation Algorithm (BOADICEA)—*BRCA1*, *BRCA2*, *PALB2*, *CHEK2* and *ATM* [11]. Following standard protocols, DNA was extracted and sequenced using the S5TM XL system (Thermo Fisher Scientific (TFS), Waltham, MA, USA), to average 200× depth. Variants with a ‘pathogenic’ or ‘likely pathogenic’ ClinVar annotation [25] and/or high-confidence predicted loss-of-function in coding regions [26] were curated following ACMG/AMP Standards and Guidelines for the Interpretation of Sequence Variants [27], including review by two or more laboratory scientists and a clinical geneticist. Analysis was restricted to single nucleotide variants and small insertions/deletions.

### 2.4. Endpoints

The study’s primary endpoint was invasive breast cancer (BC), which included incident invasive BC diagnosed during the ASPREE trial; this was adjudicated by an expert panel using histopathology, metastasis imaging or other clinical evidence [28]. Age at diagnosis of prevalent BC was self-reported as before or after 50 years.

### 2.5. Statistical Analysis

After excluding female participants with a history of prevalent BC at enrolment, multivariable Cox proportional hazards regression was used to evaluate the association between PRS with incident BC by estimating the hazard ratio (HR) per standard deviation (SD) of the PRS, after adjusting for BC family history (first-degree blood relatives), treatment (aspirin/placebo), age at enrolment, number of children, alcohol consumption, body mass index (BMI) at enrolment, and use of estrogen or estrogen/progesterone hormone replacement therapy (HRT) at enrolment. Alcohol consumption was categorised into three groupings: none (no current consumption); low (<3 drinking days per week); and high (≥3 drinking days per week). Interaction between PRS and aspirin treatment was tested independently alongside the complete set of covariates. BMI and number of children were standardised to mean = 0 (SD 1).

In a separate model, PRS was categorized into three quintile-based distribution groups—low (0–20%, Q1), moderate-risk (21–80%, Q2–4), and high-risk (81–100%, Q5). R package *pec* v2019.11.03 was used for BC-free survival prediction. Net reclassification improvement (NRI) was calculated using R package *nricens* v1.6 with 1% and 3% cutoff values for predicting increased and decreased risk categories. We calculated cumulative incidence estimates for each PRS group (multivariate adjusted), treating death as a competing risk.

Logistic regression was used to assess associations with prevalent breast cancer, including family history and the presence of rare PVs as covariates. We further stratified by diagnosis age (<50, 50+), and included number of children in the model. We used variance inflation factor (VIF) to assess the independence of predictors, and measured the discriminative ability of the PRS using concordance index and area under the receiver operating characteristic curve (AUC). Goodness-of-fit for the logistic regression was assessed using the Hosmer–Lemeshow (HL) test and the Tail-Based Max-test-statistic (TBM) [29]. DeLong’s test was used to compare between two correlated ROC curves [30,31]. We calculated PV ORs for prevalent BC for each gene and all five genes combined. Analyses were performed using R v3.6.1.

## 3. Results

### 3.1. Baseline Characteristics

The mean age of the 6339 female participants of European genetic ancestry was 75.1 years at time of enrolment, with 14% aged >80 years, and 31% current or former smokers (Table 1). The mean BMI was 28.0 kg/m^2^, 75% were current alcohol consumers and 13% had a family history of BC in a first-degree blood relative. At baseline, 533 (8.4%) participants were taking HRT (either estrogen alone or with progesterone). Six (0.2%) were taking progesterone-only preparations. Prevalent BC was reported by 475 (7.6%) participants, of which 60 (1%) were diagnosed before the age of 50 years.

### 3.2. PRS and Rare Pathogenic Variants

The PRS showed a normal distribution in the study sample with mean −0.13 (SD 0.58) before standardization (Appendix A), which was scaled to a mean of 0 (SD 1) for subsequent analyses. Forty-two rare pathogenic or likely pathogenic variants (PVs) passed our variant curation protocol (Appendix A) across 41 (0.6%) participants (one participant had two PVs detected, one each in *BRCA2* and *CHEK2*). We identified participants with PVs in the *BRCA1* (*n* = 3), *BRCA2* (*n* = 10), *PALB2* (*n* = 6), *CHEK2* (*n* = 7) and *ATM* (*n* = 16) genes. Of these participants, 20% reported a family history of BC in a first-degree blood relative at enrolment.

### 3.3. Incident Breast Cancer Risk

After excluding the 475 female participants with prevalent BC at enrolment, during median follow-up (4.7 years/participant), 110 women had incident BC. None of these women had rare PVs in the *BRCA1*, *BRCA2*, *PALB2*, *CHEK2*, and *ATM* genes. In the multivariable Cox model, conventional BC risk factors, including family history of BC, number of children, alcohol consumption and estrogen HRT were associated with risk of incident BC (Table 2).

The PRS, as a continuous variable in the same model, was an independent predictor of incident BC, with a HR of 1.43 (95% confidence interval (CI) 1.18 to 1.73, *p* < 0.001) per SD (Table 2, Figure 1), after adjustment for covariates. The PRS was also an independent predictor of incident hormone receptor (ER/PR)-positive BC (HR = 1.5 (CI 1.2–1.9), *p* < 0.001), after adjustment for covariates. The VIF for each term in the multivariable model was less than 1.1, indicating the independence of the predictors. The concordance index of the model without the PRS was 0.62 (95% CI 0.56 to 0.68), which improved after addition of the PRS to 0.65 (95% CI 0.59 to 0.71). We found no evidence of an interaction between aspirin treatment and the PRS.

We categorized the PRS into low- (Q1), moderate- (Q2–4) and high-risk (Q5) groups to consider PRS effect on incident BC. When using Q1 as a reference, participants in the high-risk PRS group had a significantly higher risk of developing incident BC compared to women in the low-risk PRS group (HR = 2.16 (95% CI 1.21 to 3.86), *p* < 0.01) (Table 2, Figure 2). The competing risk model showed that individuals in Q5 (the high-risk group) had higher cumulative incidence than those in Q1 (the low-risk group) and Q2–4 (the moderate-risk group) (Figure 2). Participants in the moderate- and low-risk groups did not have significantly different risks of incident BC. The calibration plot for the incident risk model (Appendix A) illustrates high concordance between the predicted and observed events. Net reclassification analysis had point estimates of 0.15 (95% CI 0.03; 0.24) for combined change, with NRI+ of 0.09 (95% CI −0.02; 0.13) and NRI− of 0.05 (95% CI 0.02; 0.08). Reclassification of cases and controls is shown in Appendix A.

Histopathology was available for 103 incident BC cases (Appendix A). The PRS was found to be a significant predictor of (ER+/PR+) disease (HR = 1.53 per SD (95% CI 1.22 to 1.91)), *p* < 0.001, *n* = 79).

### 3.4. Prevalent Breast Cancer

Of the 41 participants with PVs, 11 (27.5%) reported prevalent BC at baseline, compared with 7.5% (475/6339) in all female participants, giving an estimated OR of 4.69 (95% CI 2.21 to 9.27, *p* < 0.001) for PVs grouped across all five genes (Appendix A). The OR estimate for PVs in *BRCA1* and *BRCA2* (OR = 6.09 (95% CI 1.77 to 19.08)) was higher than PVs in the other genes (*ATM*/*PALB*/*CHECK2*) (OR = 3.33 (95% CI 1.19 to 7.92), *p* = 0.011). Per gene ORs are reported in Appendix A, but are limited by small carrier numbers.

In sub-group analysis, when stratifying prevalent BC cases by diagnosis age (before or after 50 years), the OR estimate for having a PV was higher for early onset BC risk (diagnosed <50 years, OR = 9.79 (CI 2.29 to 28.87), *p* = 0.02) than for later-onset BC risk (diagnosed >50 years, OR = 3.91 (CI 1.64 to 8.31), *p* = 0.02) (Appendix A).

The PRS as a continuous variable was associated with prevalent BC after controlling for covariates in the model (OR = 1.47 per SD (95% CI 1.34 to 1.61), *p* < 0.001). The HL and TBM tests did not indicate a lack of goodness-of-fit (*p* > 0.05). The AUC for the model with the PRS (AUC = 0.62 (0.59–0.65), Appendix A) was improved relative to the model including family history of BC only (AUC = 0.53 (0.52–0.55), Appendix A) (*p* < 0.01). When considering the PRS as a categorical variable, participants in the high-risk group had a significantly higher BC risk compared with the low-risk group (OR = 3.16 (95% CI 2.26 to 4.49), *p* < 0.001). Participants in the moderate-risk group also had higher BC risk versus the low-risk group (OR = 2.12 (95% CI 1.56 to 2.94), *p* < 0.001).

### 3.5. Modification of BC Risk by PRS in Rare PV Carriers

Eleven participants with PVs reported prevalent BC, and 29 with PVs reported no prevalent BC. We hypothesized that individuals with PVs and a history of BC (affected) may have a higher PRS, on average, than those with PVs but without BC (unaffected), as suggested by previous *BRCA1/BRCA2* studies [32]. However, we observed no evidence of over-representation of affected or unaffected carriers between low-, moderate- or high-risk PRS groups (Chi-squared *χ*^2^ = 1.97, df = 2, *p* = 0.37) (Figure 3, Appendix A).

## 4. Discussion

In this study, we assessed the performance of a 313-variant PRS for BC and rare pathogenic variants in BC-susceptibility genes for risk prediction of BC in older women. We found that the PRS was a significant predictor of incident BC risk, when considered as both a continuous (per SD) or categorical (low, moderate and high-risk groups) variable. Net reclassification was improved after addition of the PRS to a model composed of traditional BC risk factors. Older women in the highest quintile of the PRS distribution had over two-fold higher risk of incident BC than those in the lowest quintile. The PRS was also associated with incident hormone receptor positive (ER+/PR+) disease specifically, and with prevalent BC. The emerging clinical utility of PRS for BC risk prediction and risk stratification, previously demonstrated in women aged 40–69 years [1,2,3,4], therefore likely extends to older women.

The PRS effect (HR = 1.4 per SD) was similar to that reported in population-based studies of younger women [1,3,9,10], including a recent meta-analysis measuring the same PRS used in our study across ten prospective studies of younger women where ORs ranged from 1.48 to 1.75 across participants of all ages [1]. Subsequent validation studies of the same PRS in other cohorts found similar ORs to our study (~1.4) [2,3,4]. Notably, the average age at enrolment in the ASPREE female population (75 years) is over 15 years older than most other prior population-based studies, including the UK Biobank population, where average enrolment age was 58 years [33]. Yet, similar HRs for the PRS were observed between ASPREE and these prior studies. This suggests that the predictive performance of the PRS does not attenuate with age, in women over 70 years.

No incident BC diagnoses were observed in 41 women with rare PVs, despite the high expected BC risk conferred by these PVs (e.g., average cumulative risk to age 70 years of 50–70% for *BRCA1*/*BRCA2* PVs) [8]. This challenges the clinical value of predictive genetic testing for BC risk by sequencing of these genes alone in women aged ≥70 years. However, genetic testing for BC in older women can trigger cascade family testing, which has benefits including risk management, early detection, and/or prevention of cancer in younger family members. We also observed no incident ovarian cancer diagnoses in the 41 PV carriers.

Retrospective data suggested a higher risk of BC in individuals with a PV compared with those without (OR = 4.7) when all PVs across all genes were combined into a single group. This reflects the effect of PVs earlier in life. However, women with PVs diagnosed with cancer earlier in life are less likely to have been ascertained by our study, because they either died from cancer before the enrolment age or were too unwell to enrol in the ASPREE trial due to a current or recent cancer diagnosis. This has likely resulted the healthy selection bias that can often occur in older survivor cohorts [34]. Thus, risk estimates observed in PV carriers ascertained in our study (and the ORs and CIs reported for associations between the PRS/PVs and prevalent cases) must be interpreted with caution.

In response to recent studies reporting that an individual’s PRS may modify the penetrance of rare PVs in the *BRCA1* and *BRCA2* genes [9,10,32,35], we sought evidence of risk modification by the PRS in PV carriers affected and unaffected by prevalent BC in our study. We observed no evidence of either a higher PRS in females with a PV and affected by BC, or conversely a protective effect of a lower PRS in unaffected PV carriers (Figure 3). However, we acknowledge our analysis was limited by a relatively small number of PV carriers. Further studies are needed to investigate this more rigorously.

Clinically, it is notable that most incident BC cases in our study had favourable prognoses (e.g., hormone receptor positive). This raises the possibility that genomic risk prediction for BC in older women may have limited impact for improving survival, and that this impact must be balanced against potential overdiagnosis/overtreatment risks in this older demographic.

Key strengths of our study include the well-characterised, older study population (median age after follow-up: 78 years) followed prospectively, with all incident BC diagnoses adjudicated by an expert panel. Most previous studies of genetic risk scores for BC have examined younger cohorts, some selected for family history.

Limitations of our study include the unavailability of some phenotypic and clinical risk factors associated with BC, such as mammographic density, reproductive factors (e.g., age at menarche, menopause, first birth) and hormonal factors beyond HRT use (e.g., oral contraceptive use) [36]. Our clinical risk factor model might be improved with these additional factors. Some participants may have undergone risk-reducing bilateral prophylactic mastectomy or bilateral salpingo-oophorectomy prior to enrolment, to reduce BC/OC risk. The relatively small number of PVs detected (*n* = 41) necessitated grouping of PVs across genes to calculate averaged risk, despite known differences in the magnitude of risk conferred by PVs in different genes [12,13]. Prevalent BC (pre-enrolment) was self-reported (without specific diagnosis age) and not verified through supporting documentation, potentially causing over-estimation of prevalent BC events. Our study involved only participants of European genetic ancestry, meaning results may not be generalisable to other populations.

## 5. Conclusions

In conclusion, we demonstrate that the predictive value of a PRS for BC extends to older women, with no evidence of age-related attenuation in predictive performance after the age of 70 years. Our study has clinical implications for the use and interpretation of polygenic risk prediction of BC across the female lifespan.

## Figures and Tables

**Figure 1 cancers-13-03533-f001:**
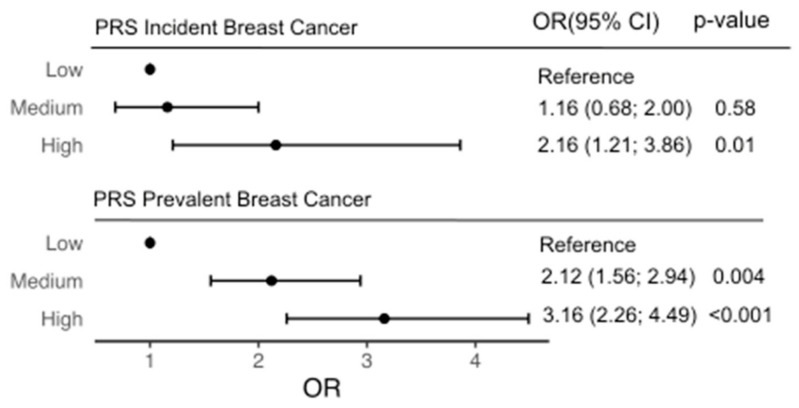
Association of a polygenic risk score (PRS) with incident and prevalent breast cancer (BC) risk in 6339 older women. We evaluated incident BC diagnoses over a median follow-up of 4.7 years and prevalent BC diagnosed pre-enrolment (self-reported). A multivariable Cox regression model including conventional risk factors examined the association between incident BC risk and the PRS as a categorical variable by quintiles (Q) of the distribution (low- (Q1), medium- (Q2–4), high- (Q5) risk groups), adjusting for family history of BC (first-degree blood relatives), treatment (aspirin/placebo), age at enrolment, number of children, alcohol consumption, body mass index (BMI) at enrolment, and use of estrogen or estrogen/progesterone hormone replacement therapy (HRT) at enrolment. Logistic regression examined associations with prevalent BC, adjusting for family history of BC (first-degree blood relatives) and pathogenic variants in BC-associated genes, represented by odds ratio (OR).

**Figure 2 cancers-13-03533-f002:**
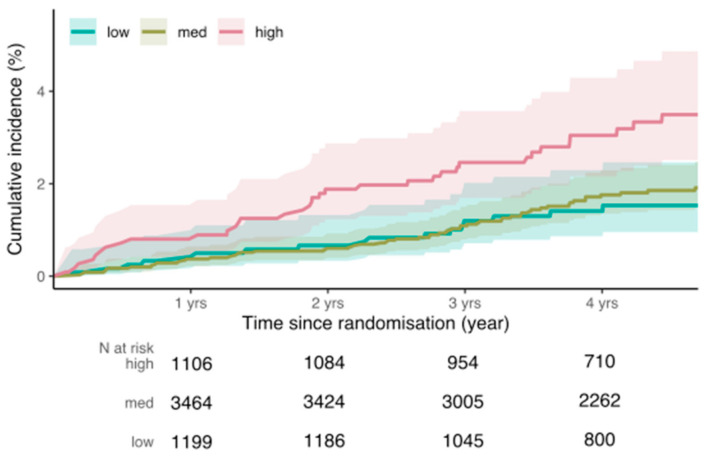
Competing risk survival curves for incident breast cancer according to PRS groups. The PRS distribution was categorized by quintiles (Q) of the distribution into three groups: low-risk (Q1, green), medium-risk (Q2–4, brown) and high-risk (Q5, red). Competing risk estimates of the cumulative incidence were calculated for each group, adjusting for the following covariates: family history of BC (first-degree blood relatives), treatment (aspirin/placebo), age at enrolment, number of children, alcohol consumption, body mass index (BMI) at enrolment, and use of estrogen or estrogen/progesterone hormone replacement therapy (HRT) at enrolment.

**Figure 3 cancers-13-03533-f003:**
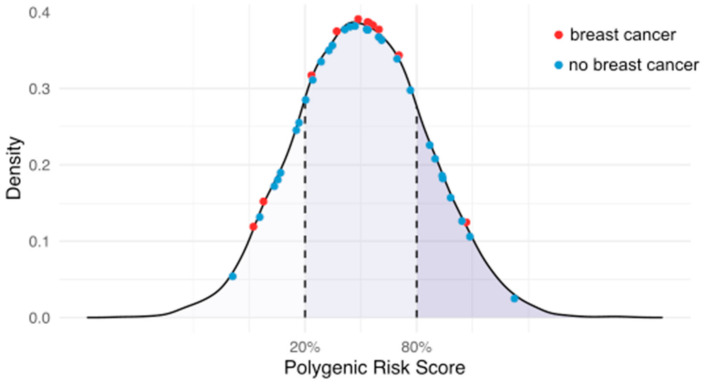
PRS distribution of female pathogenic variant (PV) carriers who were affected (red) or unaffected (blue) by prevalent BC. Density represents the proportion of individuals in each PRS group (low/medium/high). The PRS is distributed normally according to scaled PRS score (mean 0, standard deviation 1). PV carriers are highlighted across the PRS distribution.

**Table 1 cancers-13-03533-t001:** Characteristics of the study population.

Characteristics	Total	Low-Risk PRS (Q1)	Moderate-Risk PRS (Q2–4)	High-Risk PRS (Q5)
**Participants**	*n* = 6339	*n* = 1268	*n* = 3803	*n* = 1268
**Sex = Female (%)**	6339 (100)	1268 (100)	3803 (100)	1268 (100)
**Mean Age at Enrolment, Years**	75.1	75.1	75.1	75.2
**Age Group, Years (%)**				
70–74	3825 (60.3)	778 (61.4)	2287 (60.1)	760 (59.9)
75–79	1599 (25.5)	304 (24.0)	975 (25.6)	320 (25.2)
80–84	706 (11.1)	136 (10.7)	429 (11.3)	141 (11.1)
85+	211 (3.3)	50 (3.9)	112 (2.9)	47 (3.7)
**Current or Former Smoker (%)**	1974 (31.1)	391 (30.8)	1195 (31.4)	388 (30.6)
**Diabetes (%)**	497 (7.8)	97 (7.6)	289 (7.6)	111 (8.8)
**Randomized to Aspirin (%)**	3170 (50.0)	630 (49.7)	1898 (49.9)	642 (50.6)
**Body Mass Index kg/m^2^ ((mean) (SD))**	28.03 (5.09)	28.02 (5.02)	28.05 (5.10)	28.00 (5.13)
**Current Alcohol Consumption (%)**	4730 (74.6)	941 (74.2)	2850 (74.9)	939 (74.1)
**Hormone Replacement Therapy ***	533 (8.4)	103 (8.1)	321 (8.4)	109 (8.6)
**Progesterone-Only HRT**	6 (0.2)	1 (0.1)	3 (0.1)	2 (0.2)
**Family History of Breast Cancer (%) ‡**	850 (13.4)	135 (10.6)	500 (13.1)	215 (17.0)
**Prevalent Breast Cancer**				
Cases	475	47	288	140
Diagnosed <49 Years	60	6	39	15
Diagnosed 50+ Years	415	41	249	125
**Incident Breast Cancer ¶**				
Cases	129	21	66	42
**Polygenic Risk Score (mean (SD))**	0.1 (0.53)	−0.93 (0.26)	−0.13 (0.27)	0.69 (0.28)

PRS = Polygenic risk score, Q = Quintile, HRT = Hormone replacement therapy. * Estrogen alone or in combination with progesterone; ¶ Non-metastatic and metastatic events; ‡ Family history in first-degree blood relative (mother, sibling or child).

**Table 2 cancers-13-03533-t002:** Association of a polygenic risk score (PRS) with incident breast cancer (BC) risk in 6339 older women. A multivariable Cox proportional hazards regression model was used to evaluate the association between PRS as a continuous or categorical variable with incident BC (*n* = 110 cases), after adjusting for family history of BC (first-degree blood relatives), treatment (aspirin/placebo), age at enrolment, number of children, alcohol use, BMI at enrolment and use of hormone replacement therapy (HRT) at enrolment.

Variable	PRS as Continuous Variable			PRS as Categorical Variable		
	Hazard Ratio	95% CI	*p*-Value	Hazard Ratio	95% CI	*p*-Value
Polygenic Score (per standard deviation)	1.43	(1.18; 1.73)	<0.001			
Low PRS (Q1)				Reference		
Moderate PRS (Q2,3,4)				1.16	(0.68; 2.00)	0.58
High PRS (Q5)				2.16	(1.21; 3.86)	0.009
Pathogenic Variants (*n* = 41 carriers)	No incident events			No incident events		
Family History of Breast Cancer * (Y/N)	1.81	(1.15; 2.85)	0.01	1.83	(1.16; 2.88)	0.009
Age at Enrolment	0.97	(0.92; 1.02)	0.21	0.97	(0.92; 1.02)	0.22
Treatment (Aspirin)	1.16	(0.80; 1.69)	0.44	1.15	(0.79; 1.68)	0.45
Number of Children	0.81	(0.66; 0.99)	0.04	0.81	(0.66; 0.99)	0.04
Body Mass Index (kg/m^2^ (mean) SD)	1.14	(0.95; 1.37)	0.17	1.14	(0.95; 1.37)	0.15
Alcohol (None)	Reference			Reference		
Alcohol (Low)	1.16	(0.68; 1.97)	0.59	1.16	(0.68; 1.98)	0.58
Alcohol (High)	1.70	(1.01; 2.85)	0.04	1.70	(1.02; 2.86)	0.04
HRT ‡ (Y/N)	1.54	(0.88; 2.71)	0.13	1.51	(0.86; 2.65)	0.15

BMI = Body mass index, HRT = Hormone replacement therapy, PRS = Polygenic risk score, CI = Confidence Interval, SD = Standard deviation. * Family history in first-degree blood relative (mother, sibling or child); ‡ Estrogen alone or in combination with progesterone.

## Data Availability

The data that support the findings of this study are available from the corresponding author upon request.

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
