# Peer review of "Genomic Risk Prediction for Breast Cancer in Older Women"

_cancers, 2021, doi:10.3390/cancers13143533_

Round 1
Reviewer 1 Report
Lacaze and coworkers have assessed the effect of PRS on incident breast cancer in the older European ancestry women in a clear and concise way. They show the association of 313-variant PRS with overall and ER+/PR+ BC risk. They have also rightly mentioned the limitations of their study wrt to generalizability, healthy selection bias and limited availability of important covariates
However there are minor issues that should be addressed for further clarity:
- The authors present the estimates for ER+/PR+ BCs in abstract and manuscript text. However, this also needs to be presented as main results. Whether the estimates for ER+/PR+ also adjusted for family history of BC, treatment, age at enrolment, number of children, alcohol use, BMI and use of hormone replacement therapy is not clear.
- The tables are not self-explanatory. It is not clear which confounders were adjusted for in most of the tables and figures (Fig1, Fig 2, Table S4, S5, S6). Particularly were the ORs presented in Figure 1 for prevalent BCs adjusted for the confounders? The list of confounders should be stated as footnotes below respective tables and figures.
- The presentation of Table S4 is not very clear. Was a combined effect of PVs and PRS were observed or are the estimates 4.69 (2.21 – 9.27) and 4.64 (2.19-9.15) adjusted for PRS
Author Response
We thank the Reviewers for their comments and careful consideration of our study. Please find below our responses to each reviewer point. We now provide the revised article, which has been strengthened by peer review.
Review 1 Report
Comments and Suggestions for Authors
Lacaze and coworkers have assessed the effect of PRS on incident breast cancer in the older European ancestry women in a clear and concise way. They show the association of 313-variant PRS with overall and ER+/PR+ BC risk. They have also rightly mentioned the limitations of their study wrt to generalizability, healthy selection bias and limited availability of important covariates
However there are minor issues that should be addressed for further clarity:
The authors present the estimates for ER+/PR+ BCs in abstract and manuscript text. However, this also needs to be presented as main results.
- Thank you, we have now added this to the main results (line 178)
Whether the estimates for ER+/PR+ also adjusted for family history of BC, treatment, age at enrolment, number of children, alcohol use, BMI and use of hormone replacement therapy is not clear.
- Yes, similar to the base model for incident BC, we confirm that the estimates for ER+/PR+ BC are adjusted for the following covariates (family history of BC [first-degree blood relatives], treatment [aspirin/placebo], age at enrolment, number of children, alcohol consumption, body mass index [BMI] at enrolment, and use of estrogen or estrogen/progesterone hormone replacement therapy [HRT] at enrolment). We have now added text to the Results section to clarify this point.
The tables are not self-explanatory. It is not clear which confounders were adjusted for in most of the tables and figures (Fig1, Fig 2, Table S4, S5, S6). Particularly were the ORs presented in Figure 1 for prevalent BCs adjusted for the confounders? The list of confounders should be stated as footnotes below respective tables and figures.
- As requested, we have now added a list of the confounders/covariates used for adjustment in the models, to the following Figures and Tables.
- Figure 1, Figure 2
- Table 2
- Tables S4, S5, S6
- Please note the list of confounders for Table S4, S5, S6 (prevalent BC analysis) was limited, because we only have limited information for prevalent breast cancer.
The presentation of Table S4 is not very clear. Was a combined effect of PVs and PRS were observed or are the estimates 4.69 (2.21 – 9.27) and 4.64 (2.19-9.15) adjusted for PRS
- We confirm that the estimates reported for PVs (4.69 [2.21 – 9.27]) were adjusted for PRS. We have now added a comment to Table S4 accordingly.

Reviewer 2 Report
This is an interesting study asking an important question - whether PRS and PV testing for breast cancer risk is clinically relevant in later life. The PRS results are interesting, suggesting that PRS remains predictive at older age. Unfortunately the numbers of PVs carriers are too small to comment.
General comments
Part of this paper concerns the effect of rare PVs in older age. It would be good to reference in the introduction recent studies that have shown the increase in breast cancer risk associated with PTVs in the high risk genes (BRCA1, BRCA2, etc) decreases with increasing age.
The analysis of prevalent BC is confusing and doesn't add anything to the paper. The main subject of this paper is risk in women over 70 years, so comparing past diagnoses before and after 50 years adds noise. In addition, the association between PVs and breast cancer has been well researched in much larger, prospective samples. I wonder if the analysis of prevalent BC was added because no PVs were identified among incident cases. While that is a shame, it does not seem to be a good reason to add the prevalent BC analysis. However, the fact no PVs were observed in the incident cases is, in itself, important to know and the comments around this issue in the discussion are worth keeping.
It is also important to know that 475 of the "unaffected" participants in the incident BC analysis had (prevalent) BC previously - the PRS analysis for incident breast cancer should be repeated after excluding the prevalent cases.
For the discussion/conclusions, to improve the clinical relevance of this paper, do you have any suggestion of what could be the suggested clinical action for over 70s identified at higher risk?
Specific comments:
In the introduction, I recommend moving the sentence quoting 18% familial risk accounted for by PRS (page 2, beginning line 62) to the first paragraph which discusses PRS (its currently in the second paragraph which is focused on rare PVs).
Page 2, line 95: fix reference (1)
Table 1: the numbers in the row for progesterone-only HRT don't seem to add up
In supplementary material, please provide list of 313 variants indicating the 271 that were included (and their effect size as used in the PRS) and those that were not found/excluded during QC.
Methods: was there any adjustment for country (Australia vs US)?
Page 6, line 196: do you mean Figure 3?
For individual tests, please quote specific p-values, not < some threshold e.g. <0.01.
Supplementary Figures 2 b, c, d - relate to different analyses (incident vs prevalent) please clarify in legend or split into different figures (or remove prevalent, see earlier comment)
Why does the conclusions title appear after the conclusions written in the text? - Editorial team response: This is to be added as part of our editorial revision process, thank you.
Author Response
Comments and Suggestions for Authors
This is an interesting study asking an important question - whether PRS and PV testing for breast cancer risk is clinically relevant in later life. The PRS results are interesting, suggesting that PRS remains predictive at older age. Unfortunately the numbers of PVs carriers are too small to comment.
General comments
Part of this paper concerns the effect of rare PVs in older age. It would be good to reference in the introduction recent studies that have shown the increase in breast cancer risk associated with PTVs in the high-risk genes (BRCA1, BRCA2, etc) decreases with increasing age.
- We thank the Reviewer for this comment. We have now added a sentence to the Introduction regarding evidence for decreasing risk with age, citing the following 5 studies of high-risk BC genes, involving large sample sizes and age ranges (line 61):
- Prevalence and penetrance of BRCA1 and BRCA2 mutations in a population-based series of breast cancer cases. Anglian Breast Cancer Study Group. Br J Cancer. 2000;83(10):1301-1308.
- Kuchenbaecker KB, Hopper JL, Barnes DR, et al. Risks of Breast, Ovarian, and Contralateral Breast Cancer for BRCA1 and BRCA2 Mutation Carriers. JAMA. 2017;317(23):2402-2416.
- Mavaddat N, Peock S, Frost D, et al. Cancer risks for BRCA1 and BRCA2 mutation carriers: results from prospective analysis of EMBRACE. J Natl Cancer Inst. 2013;105(11):812-822.
- Breast Cancer Association C, Dorling L, Carvalho S, et al. Breast Cancer Risk Genes - Association Analysis in More than 113,000 Women. N Engl J Med. 2021;384(5):428-439.
- Hu C, Hart SN, Gnanaolivu R, et al. A Population-Based Study of Genes Previously Implicated in Breast Cancer. N Engl J Med. 2021;384(5):440-451.
The analysis of prevalent BC is confusing and doesn't add anything to the paper. The main subject of this paper is risk in women over 70 years, so comparing past diagnoses before and after 50 years adds noise. In addition, the association between PVs and breast cancer has been well researched in much larger, prospective samples. I wonder if the analysis of prevalent BC was added because no PVs were identified among incident cases. While that is a shame, it does not seem to be a good reason to add the prevalent BC analysis. However, the fact no PVs were observed in the incident cases is, in itself, important to know and the comments around this issue in the discussion are worth keeping.
- The prevalent BC analysis was included in the study because, a) there was a high number of prevalence BC cases observed in the population (N=475 events) and b) there was a significant association observed between prevalent BC and the PRS (OR=1.47 per SD, [1.34-1.61], p<0.001) and PVs (OR=4.69 [2.21-9.27], p<0.001). However, we acknowledge the various issues, caveats and limitations with prevalent BC analysis, and we have subsequently de-emphasised the prevalent BC results in our manuscript accordingly, for the reasons highlighted by Reviewer 2.
It is also important to know that 475 of the "unaffected" participants in the incident BC analysis had (prevalent) BC previously - the PRS analysis for incident breast cancer should be repeated after excluding the prevalent cases.
- We can confirm that we already excluded all participants with prevalent BC from the incident analysis. The 475 female participants with prevalence BC at enrolment were not included in any of the incident BC analyses. We have now added sentences to the Methods (line 118) and Results (line 168) sections to clarify this point.
For the discussion/conclusions, to improve the clinical relevance of this paper, do you have any suggestion of what could be the suggested clinical action for over 70s identified at higher risk?
- After discussions with our clinical colleagues, we are somewhat reluctant to suggest (generalize) clinical actions for BC risk to all older women, based on the results of our study alone. BC risk assessment is complex, involving use of many different risk factors (clinical and genetic) - some of which were not available for our study - such as mammographic density, reproductive factors (e.g. age at menarche, menopause, first birth) and hormonal factors beyond HRT use (e.g. oral contraceptive use)). Interventions following BC risk assessment in older women also need to be balanced with the potential for overdiagnosis and overtreatment in this age group. We also acknowledge the lack of ethnic diversity among participants in our study, further limiting generalizability. Therefore, we have instead chosen to use careful statements regarding clinical implications in the Discussion, such as:
“The emerging clinical utility of PRS for BC risk prediction and risk stratification, previously demonstrated in women aged 40-69 years, therefore likely extends to older women”.
“Clinically, it is notable that most incident BC cases in our study had favourable prognoses (e.g. hormone receptor positive). This raises the possibility that genomic risk prediction for BC in older women may have limited impact for improving survival, and that this must be balanced against overdiagnosis/overtreatment risks in this older demographic.”
Specific comments:
In the introduction, I recommend moving the sentence quoting 18% familial risk accounted for by PRS (page 2, beginning line 62) to the first paragraph which discusses PRS (its currently in the second paragraph which is focused on rare PVs).
- We thank the reviewer for this comment and have now made this change.
Page 2, line 95: fix reference (1)
- We have now made this change.
Table 1: the numbers in the row for progesterone-only HRT don't seem to add up
- We have now made this change.
In supplementary material, please provide list of 313 variants indicating the 271 that were included (and their effect size as used in the PRS) and those that were not found/excluded during QC.
- We have now provided this Table in the Supplementary Materials (Table S8).
Methods: was there any adjustment for country (Australia vs US)?
- We confirm that the final analysis for this study was conducted on 6339 female ASPREE participants from Australia aged ≥70 years, for whom both genome-wide genotyping and targeted sequencing data was available (line 89). Participants from the US were not included due to the unavailability of targeted sequencing data for US participants - therefore no adjustment for country was made.
Page 6, line 196: do you mean Figure 3?
- This has now been corrected
For individual tests, please quote specific p-values, not < some threshold e.g. <0.01.
- After consultation with biostatistician colleagues, and in accordance with the journal formatting requirements, it was recommended to keep our original p-value notations.
Supplementary Figures 2 b, c, d - relate to different analyses (incident vs prevalent) please clarify in legend or split into different figures (or remove prevalent, see earlier comment)
- We have now clarified this in the figure legend as requested
Why does the conclusions title appear after the conclusions written in the text? - Editorial team response: This is to be added as part of our editorial revision process, thank you.
- This has now been resolved
Round 2
Reviewer 2 Report
Thank you for your responses. The manuscript reads better now. Just to note, the numbers in the table for progesterone-only HRT now add up but the number in the text (Thirteen) is still incorrect.